# Evaluation of a library of *loxP* variants with a wide range of recombination efficiencies by Cre

Yuji Yamauchi[1,2], Hidenori Matsukura[1], Keisuke Motone[3], Mitsuyoshi Ueda[1], Wataru Aoki[1]*

1 Division of Applied Life Sciences, Graduate School of Agriculture, Kyoto University, Sakyo-ku, Kyoto, Japan, 2 Japan Society for the Promotion of Science, Sakyo-ku, Kyoto, Japan, 3 Paul G. Allen School of Computer Science and Engineering, University of Washington, Seattle, Washington, United States of America

* aoki.wataru.6a@kyoto-u.ac.jp

**Data Availability Statement:** All relevant data are within the manuscript and its Supporting Information files except row Illumina sequencing data. Row Illumina sequencing data files are

## Abstract

Sparse labeling of individual cells is an important approach in neuroscience and many other fields of research. Various methods have been developed to sparsely label only a small population of cells; however, there is no simple and reproducible strategy for managing the probability of sparse labeling at desired levels. Here, we aimed to develop a novel methodology based on the Cre-*lox* system to regulate sparseness at desired levels, and we purely analyzed cleavage efficiencies of *loxP* mutants by Cre. We hypothesized that mutations in the *loxP* sequence reduce the recognition efficiency by Cre, which enables the regulation of the sparseness level of gene expression. In this research, we mutagenized the *loxP* sequence and analyzed a library of *loxP* variants. We evaluated more than 1000 mutant *loxP* sequences, including mutants with reduced excision efficiencies by Cre ranging from 0.51% to 59%. This result suggests that these mutant *loxP* sequences can be useful in regulating the sparseness of genetic labeling at desired levels.

## Introduction

Sparse labeling is a genetic methodology that is used to label only a small number of cells in an overall population. Sparse labeling has impacted a variety of fields, and is especially important in neuroscience, because a massive diversity of neurons with unique morphologies is present in the nervous system [1, 2] and a tremendous number of neuronal cells exist in the brain; approximately 86 billion in the human brain and 100 million in the mouse brain [3, 4]. In addition, brains are tightly packed with neurons, and their mixed dendritic and axonal processes hamper the visualization of distinct morphologies. The paradigm in which one examines the characteristics of stochastically selected subsets of cells of the same type is particularly useful because it enables single-cell analysis to elucidate the functional logic of neuronal circuits. In this context, there is a great demand for stochastic gene expression for small populations of cells.

available from the NCBI database (NCBI SRA accession: SRR19749293).

**Funding:** Y.Y. Grant No. 20J22603 KAKENHI, Japan Society for the Promotion of Science, https://www.jsps.go.jp/j-grantsinaid/20_tokushourei/index.html W.A. Grant No. JPMJPR16F1 Precursory research for embryonic science and technology, Japan Science and Technology, Japan https://www.jst.go.jp/kisoken/presto/index.html The funders had no role in study design, data collection and analysis, decision to publish, or preparation of the manuscript.

**Competing interests:** The authors have declared that no competing interests exist.

Several methodologies have been developed to achieve sparse labeling. One method consists in the use of an animal line with the desired expression patterns after screening transgenic lines with variegated gene expression [5–8]. Other methods have relied on site-specific recombinases (SSRs). In the methods using SSRs, the sparseness level can be regulated in several ways. First, the sparseness level can be regulated by a suitable tamoxifen dosage in a CreER-*lox*-mediated recombination system [9–15]. Second, viral injections of a low titer into Cre driver lines can be implemented [16, 17]. However, these methodologies have major problems in that it is difficult to determine the sparseness levels a priori with reproducibility, because they require very sophisticated experimental techniques or the time-consuming titration of chemical or gene induction conditions to limit the spatial and/or temporal expression of a recombinase.

Methodologies that can regulate sparseness at a predicted level with high reproducibility would be highly useful. In this situation, several methods have been developed in recent years. First, the mononucleotide repeat frameshift (MORF) method is a Cre-dependent sparse cell labeling approach based on mononucleotide repeat frameshifting as a stochastic translational switch. MORF can regulate sparseness levels with high reproducibility [18, 19]. The labeling rate is approximately ~1%– 5%, depending on the targeted cell type and Cre line used. Second, the mosaic analysis with a repressible cell marker (MARCM) and mosaic analysis with double markers (MADM) transgenic approaches have been established to sparsely label cells based on Cre-*lox*-mediated interchromosomal recombination, which occurs during mitosis [20–22]. The labeling rate is approximately ~1%– 5%, depending on the targeted cell type and Cre line used. These methods can label effectors with high reproducibility; however, the labeling rate is dependent on cell type and Cre line and cannot tune sparseness to a desired level. In addition, MADM and MARCM can be used only in cells undergoing mitosis. Third, sparse predictive activity through recombinase competition (SPARC) is a method that utilizes PhiC31 recombinase and two competing *attP* target sequences and an *attB* target sequence [23]. SPARC used three types of progressively truncated *attP* sequences to regulate the sparseness level. SPARC-D (Dense) labeled 48%– 51% of cells, SPARC-I (Intermediate) labeled 17%– 22% of cells, and SPARC-S (Sparse) labeled 3%– 7% of cells. Finally, stochastic gene activation with genetically regulated sparseness (STARS) [24, 25] was derived from Brainbow [26]. The Brainbow system is a method to stochastically label cells using two mutually exclusive *lox* sequences. STARS regulates the sparseness level of the effector by lengthening a spacer DNA sequence between *lox2272* sequences, to regulate the excision efficiency by Cre. STARS transgenes that possess various lengths of spacers can regulate the sparseness level from 5% to 50% of the cell population. These methods yielded highly reproducible sparse labeling; however, it remains difficult to regulate the sparseness rate at desired levels. For example, SPARC can only adjust the sparseness level by three levels. STARS require a very long spacer DNA (e.g., > 10 kb) to achieve low stochastic labeling rate, which hampers the construction of transformants.

To overcome the difficulties of low reproducibility and low regulatability reported previously, we sought to develop a novel methodology that allows achieving sparseness at desired levels with high reproducibility by adapting the widely used Cre-*lox* recombination system [27–31]. In particular, we focused on the Brainbow system mentioned above. We hypothesized that the expression rate of a gene could be regulated by introducing a mutation in one of the *lox* sequences (*lox2272* or *loxP*) to reduce the recognition rate by Cre [32–43]. We performed random mutagenesis on the *loxP* sequence, and obtained mutants with reduced excision rates by Cre. The *loxP* mutants acquired in this study could be used to label genes in a simple and reproducible way and potentially regulate gene expression in cell populations with a desired level of sparseness. This method is likely to be particularly effective because Cre can be applied to a wide range of organisms, including mice, flies, and worms, and can be employed in post-mitotic cells [44–48].

# Results

## Strategy for achieving recombination efficiency at the desired rate

Cre-mediated recombination occurs between one of the two identical pairs of *lox* sites (a pair of *loxP* sequences and a pair of *lox2272* sequences) in a mutually exclusive way in the Brainbow system [26]. Moreover, excision by one recombination event removes a *lox* site, which is required for the other recombination event to occur. In the genetic circuit in which two pairs of *lox* sequences are inserted alternately and gene A is inserted between *loxP* sequences and gene B is inserted between *lox2272* sequences, the decision between the expression of gene A and gene B becomes stochastic and mutually exclusive (**Fig 1A**). The recognition efficiency of two pairs of *lox* sequences (*loxP* sequences and *lox2272* sequences) is comparable [26]. Thus, the expression rates of gene A and gene B are expected to be approximately the same. To develop a method that can stochastically activate gene expression with a desired sparseness level, we first considered the reaction kinetics of Cre-*lox*-mediated intrachromosomal recombination. We hypothesized that the affinity of Cre for the mutagenized *loxP* sequence could be reduced relative to that for the *lox2272* sequence (**Fig 1B**) and that we could regulate the expression rate of gene A and gene B using the mutagenized *loxP* sequences.

## Strategies for the construction and evaluation of a library of mutant *loxP* sequences

We designed a strategy to analyze the effect of a mutation in the *loxP* sequences on the excision rate by Cre in a high-throughput manner (**Fig 2**). First, we mutagenized one of the recombinase binding elements (RBEs) of the *loxP* sequence by PCR and constructed a library of mutant *loxP* sequences (**Fig 2A and 2B**). Next, the library was cloned into a centromere-type plasmid and introduced into *Saccharomyces cerevisiae with a CreEBD system, which is a* β-estradiol-inducible Cre expression construct [49]. Cre was then induced in yeast with the mutant *loxP* library (**Fig 2C**). After Cre induction, the library of mutant *loxP* sequences was extracted from the yeast (**Fig 2D**). Subsequently, the library of mutant *loxP* sequences was subjected to Illumina sequencing and the cleavage rate between *loxP* sequences was quantified (**Fig 2E**). Finally, to verify the accuracy of the Illumina sequencing results, we randomly selected *loxP* variants and quantified their cleavage rates by quantitative polymerase chain reaction (qPCR) (**Fig 2F**).

## Design of a library of mutant *loxP* sequences

In Cre-*lox* recombination, Cre forms a complex with *lox* sequences by recognizing inverted repeats consisting of 13 bp on each side of the *lox* sequences, named RBEs [31]. In our study, we mutagenized 13 bp (5′- ATAACTTCGTATA-3′) of the right RBE of the *loxP* sequence. We predicted that an increase in the number of substitutions would result in a reduction of the affinity of *loxP* variants for Cre. Regarding the substitutions of 3 or more bases, it was difficult to obtain sufficient coverage rate because of the exponentially increasing number of combinations. Thus, the goal of this research was to evaluate most mutants with one or two nucleotide substitutions and a fraction of the mutants with substitutions of three or more nucleotides (**Fig 3A**). The right RBE of the *loxP* sequence was designed to preserve the original base at 84.7% probability, to obtain as many 2-base substitutions as possible. For example, if the original base was A, it was designed so that 84.7% remained as A, 5.1% as T, 5.1% as G, 5.1% as C. When the mutated rate in randomized primer is 15.3%, 2-base substitutions are most efficiently obtained. In this case, about 30% of all mutants of *loxP* sequences would have 2 base substitutions (**Fig 3B**). The RBE sequences of 30 samples after mutagenesis by PCR were

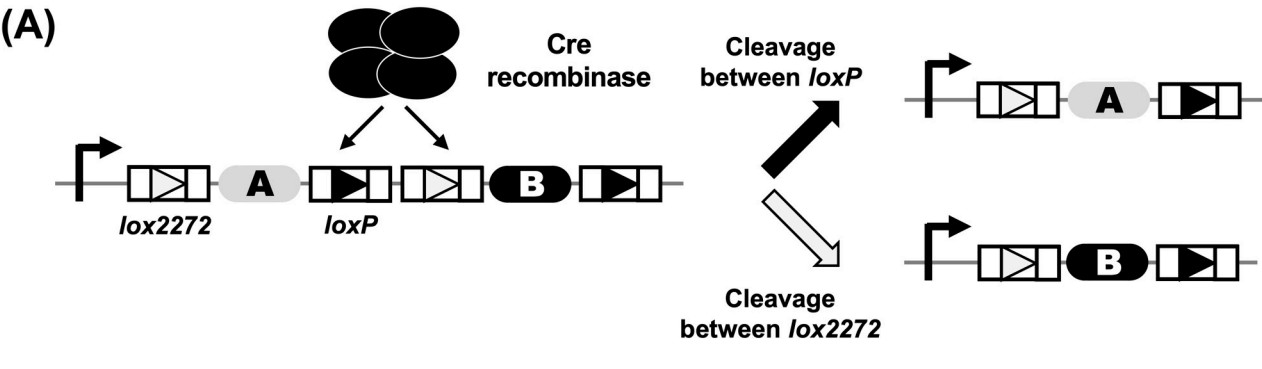

**Fig 1. Schematic overview of our strategy. (A)** Stochastic excision of *lox* sequences by Cre. Gene A is excised when Cre cleaves *lox2272* sequences and gene B is excised when Cre cleaves *loxP* sequences. Cre cleaves one of two pairs of *lox* sequences (a pair of *lox2272* sequences or a pair of *loxP* sequences) in a mutually exclusive way, which leads to the exclusive expression of gene A or gene B. **(B)** Regulation of the sparse labeling rate using *loxP* variants. We hypothesized that mutations in *loxP* sequences affect the affinity of the *loxP* sequence against Cre, thus rendering them less likely to be cleaved by Cre.

confirmed by Sanger sequencing, and the mutations were successfully introduced at the desired position (**S1 Table**). As a result of the comparison of Sanger sequencing and simulation, we obtained the library of *loxP* variants with approximately the expected rate of the number of substitutions (**Fig 3B**).

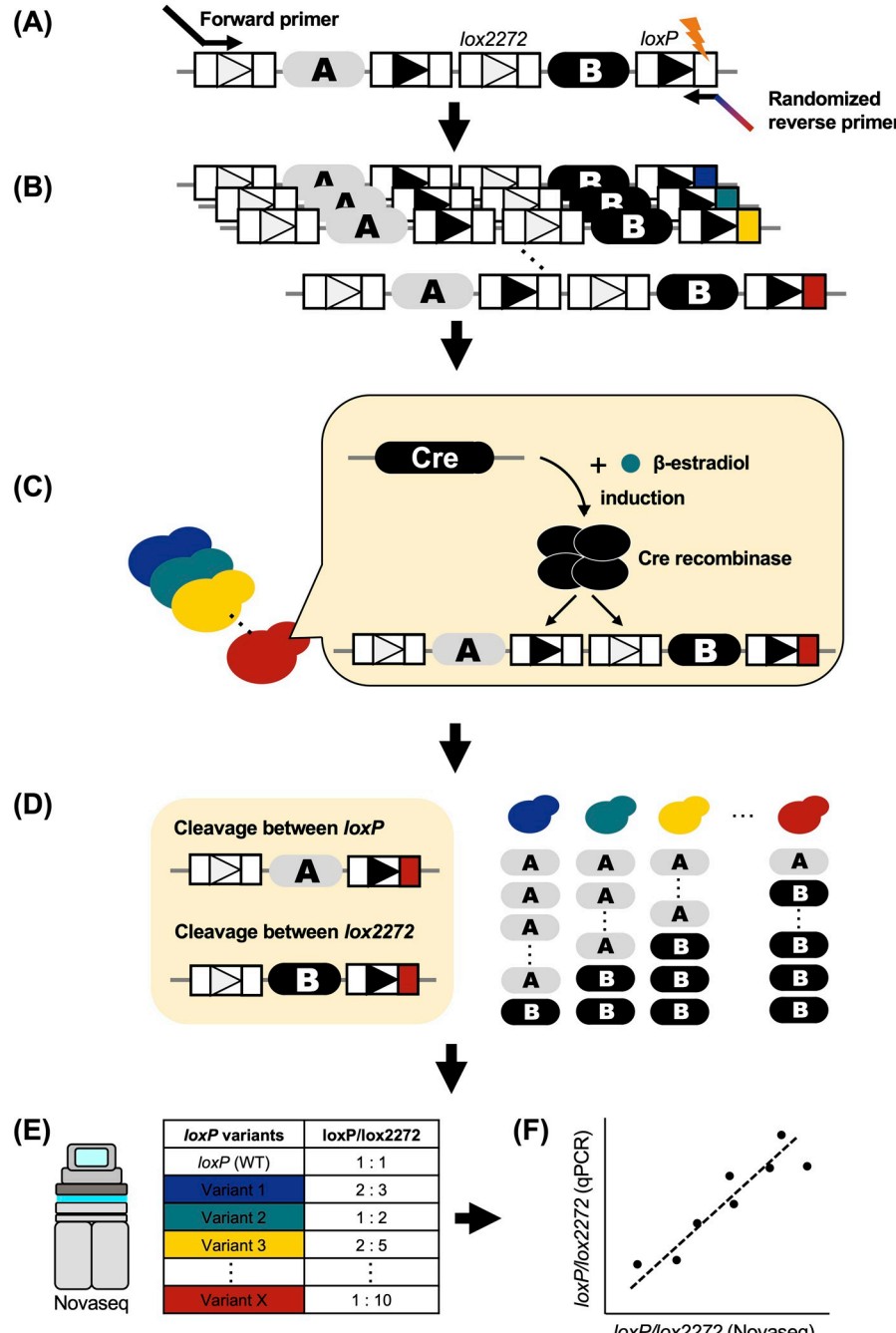

**Fig 2. Workflow of this research.** Our workflow was divided into six steps. **(A,B) Mutagenesis by PCR.** We mutagenized one of the RBEs of the *loxP* sequence and established a library of mutant *loxP* sequences. **(C) Transformation and Cre induction.** The library of mutant *loxP* sequences was transformed into Saccharomyces cerevisiae and Cre was induced. **(D) Extraction.** After Cre induction, we extracted a library of mutant *loxP* sequences from S. cerevisiae. **(E) Analysis.** We analyzed the excision rate of the mutant *loxP* sequences by Novaseq. **(F) Validation.** We quantified the excision rate of *loxP* variants by qPCR, to validate the results of the Illumina sequencing analysis.

## Analysis of the library of mutant *loxP* sequences by Illumina sequencing

The library of mutagenized *loxP* sequences was analyzed by Illumina sequencing after a cleavage reaction that was carried out by inducing Cre using the CreEBD system in yeast. The

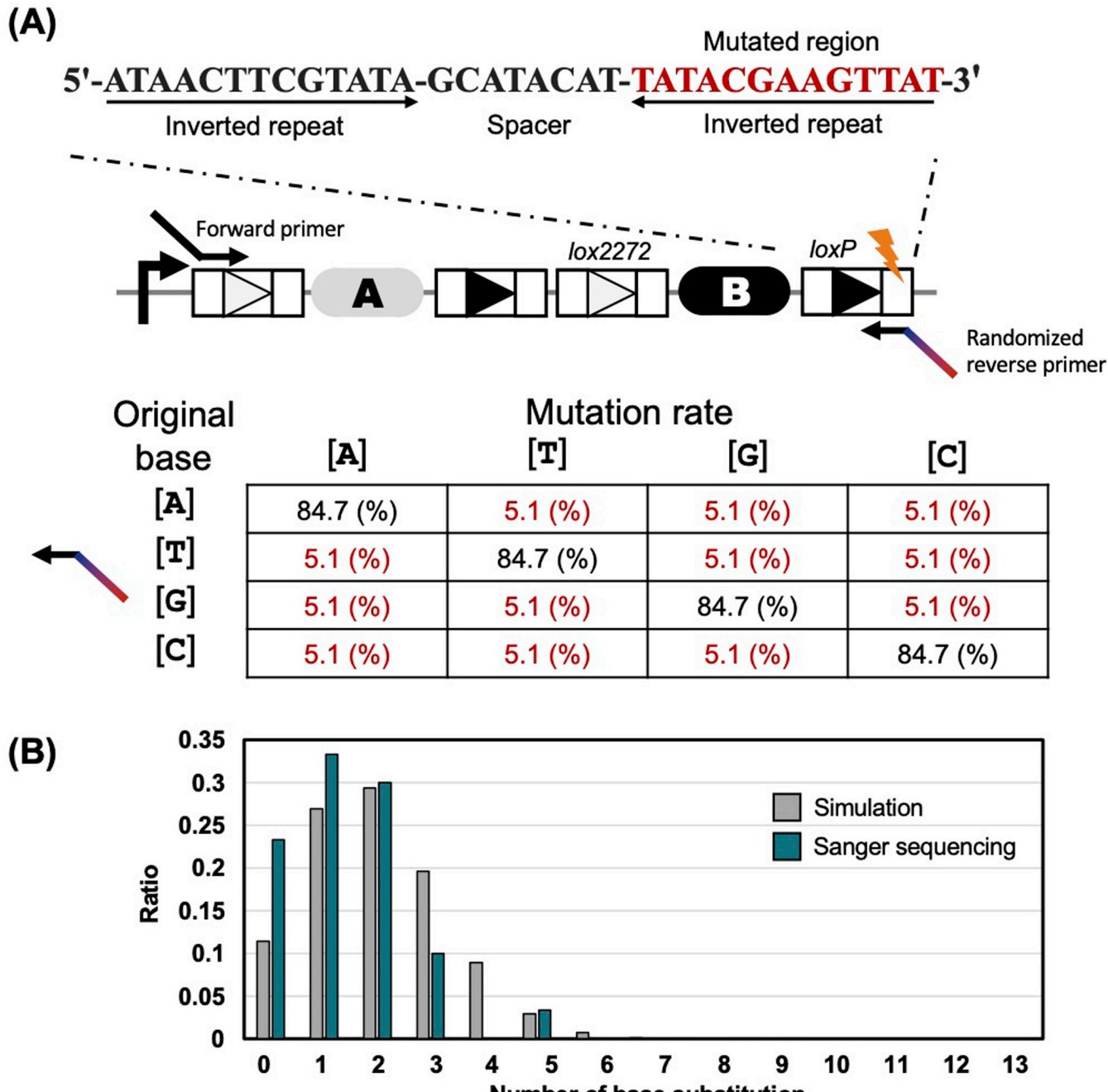

**Fig 3. Primer design to construct a library of *loxP* variants.** (A) Primer design to mutagenize the *loxP* sequence. We introduced a mutation into the right arm (13 bp) of the *loxP* sequence. The right arm of the *loxP* sequence is shown in red. We set the mutation rate of the primer to 15.3% to obtain as many 2-base substitutions as possible. When the mutated rate in randomized primer is 15.3%, 2-base substitutions are most efficiently obtained. (B) Validation of the distribution of the number of base substitutions by Sanger sequencing. Gray indicates the theoretical value (mutation rate = 15.3%). Green indicates the results of Sanger sequencing. The detailed results of Sanger sequencing are provided in **S1 Table**.

workflow of this analysis is shown in **Fig 4A**. First, Illumina sequencing reads were classified according to the type of mutation of the *loxP* sequences (**Fig 4A1 and 4A2**). In total, we generated 16471 different mutants of *loxP* sequences. To obtain accurate data, we focused on 1111 *loxP* variants with a number of Illumina sequencing reads greater than 500 (**S1 File**). Next, we

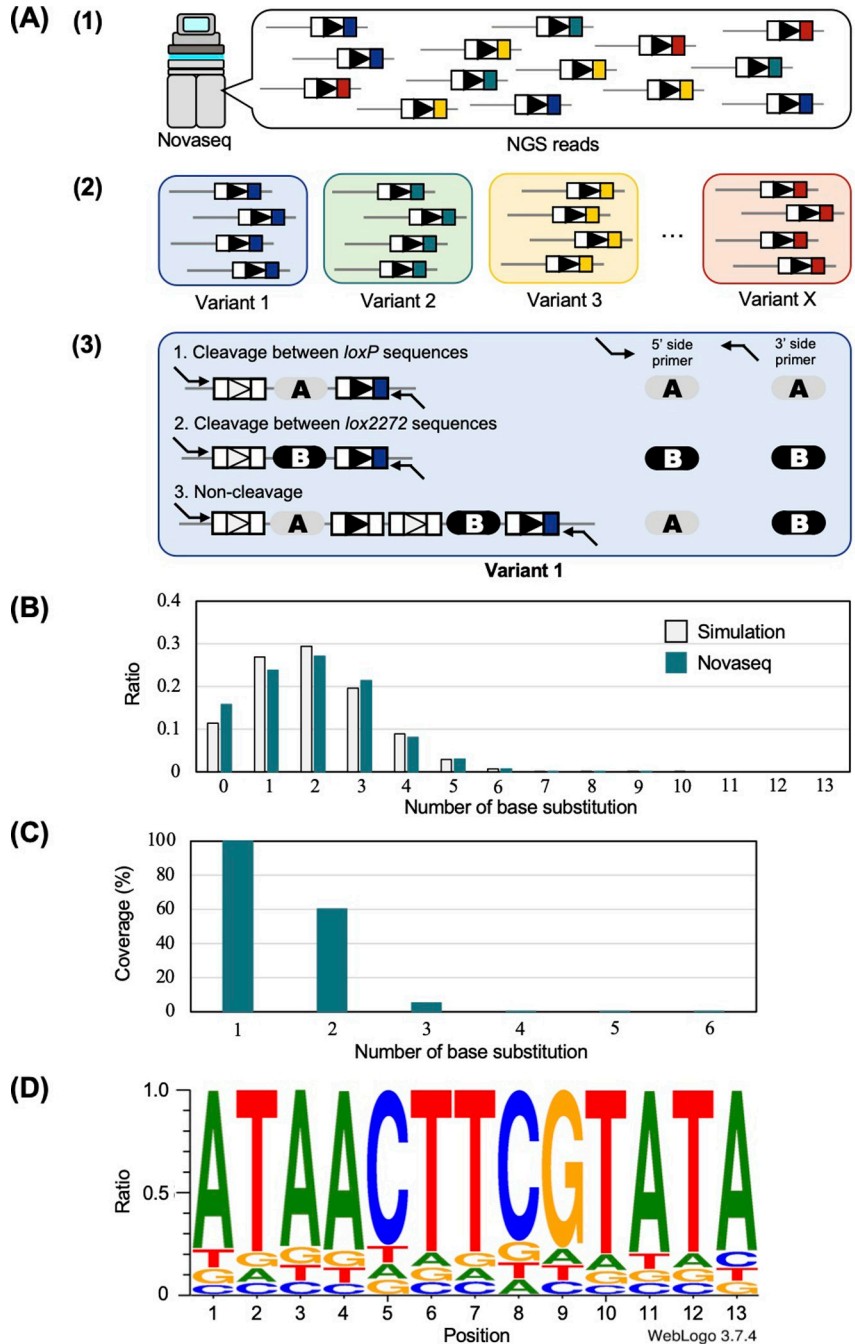

**Fig 4. Analysis of the cleavage rate of *loxP* variants by Novaseq. (A)** Schematic overview of the Illumina sequencing analysis. We analyzed the Illumina sequencing data in three steps. (1) We extracted DNA sequences that possessed sequences of *loxP* variants. (2) All paired-end Illumina sequencing reads were classified according to the type of mutation in *loxP* sequences. (3) Using the paired-end Illumina sequencing reads obtained using the primer at the 5′ side and the primer at the 3′ side, we identified the patterns of recombination of Illumina sequencing reads for each *loxP* variant. If an Illumina sequencing read possessed sequence A, we determined that the plasmid was cleaved between *loxP* sequences by Cre. If an Illumina sequencing read possessed sequence B, we determined that the plasmid was cleaved between *lox2272* sequences by Cre. If an Illumina sequencing read possessed sequences A and B, we determined that the plasmid was not cleaved. **(B)** Comparison of the distribution of the number of substituted bases between experimental values and theoretical values. The light-gray color indicates the theoretical values (substitution rate = 15.3%). Green indicates the experimental value. This method of calculation is provided in the Materials and Methods. **(C)** Coverage rate (%) of substitution. **(D)** Nucleotide rate at each position of *loxP* sequences.

analyzed the cleavage rate of individual *loxP* mutant sequences using the combination of the Illumina sequencing reads acquired using a 5′ primer with those acquired using a 3′ primer (**Fig 4A3**). As a result of the calculation, the average non-cleavage rate was 3.5%, which confirmed that Cre was correctly induced. Before performing a detailed analysis of individual mutant *loxP* sequences, we confirmed the quality of the library of mutagenized *loxP* sequences. First, the distribution pattern of the number of base substitutions roughly corresponded to the theoretical one (**Fig 4B**), as previously suggested in **Fig 3B** using Sanger sequencing. The coverage rate of base substitution was examined. As a result, we acquired 100% (39/39 variants) of the single-base substitutions and 60.5% (425/702 variants) of the 2-base substitutions (**Fig 4C**). Third, we assessed the nucleotide bias at every 13 positions of the right RBE of the *loxP* sequence. The results showed that no bias existed in the type of bases that were introduced into each of the positions of the RBE (**Fig 4D**).

Subsequently, we quantified the cleavage rate of individual *loxP* sequences. As predicted, as the number of substitutions increased from a 1-base substitution to 6-base substitutions, the average cleavage rate between *loxP* sequences decreased monotonically (**Fig 5A**). As shown in **Fig 5B**, we obtained mutant *loxP* sequences with reduced recognition efficiencies by Cre in various proportions (**Table 1**). These results support our hypothesis that sparseness can be regulated by mutagenizing an RBE of *lox* sequences.

## Quantification of the cleavage rate of the mutant *loxP* sequences by qPCR

We assessed the cleavage rate of the mutagenized *loxP* sequences by qPCR to validate the accuracy of the results of the Illumina sequencing analysis. The results indicated that the addition of various mutations to the RBE can alter the cleavage efficiency of Cre at various rates. However, the sequencing results potentially have a certain bias because it requires PCR during sample preparation. To confirm that there is no significant bias in the sequencing results, we performed qPCR, which is low-throughput but can accurately quantify the cleavage rates by Cre. We quantified the non-cleavage rate and *loxP* cleavage rate in each mutant *loxP* sequence by qPCR. A comparison of non-cleavage rates in the absence or presence of Cre induction showed that the recombination events that occurred were dependent on Cre induction (**Fig 6A**). The cleavage rates of the randomly selected *loxP* variants were quantified using qPCR and compared with the results of the Illumina sequencing analysis (**Fig 6B** and **S2 Table**). These results showed that the cleavage rate between *loxP* sequences was highly correlated with that obtained by Illumina sequencing analysis ($R^2 = 0.9695$) (**Fig 6C**). This qPCR result showed that the Illumina sequencing data are reliable.

## Identification of mutant *loxP* sequences with less than 1% recognition efficiency

Although we analyzed 1111 variants and successfully identified variants with a recognition efficiency ranging from 17% to 59% (**Fig 5B**), the sparse labeling method requires mutant *loxP* sequences with a lower cleavage efficiency as low as 1%. Fig 5A indicates that the recognition rate by Cre drops as the number of base substitutions increase. To obtain mutant *loxP* sequences with lower recognition rates by Cre, we evaluated the recognition rate of mutant *loxP* sequences with more than 7-base substitutions. As a result of qPCR quantification, we obtain several mutant *loxP* sequences with recognition efficiencies of less than 1% (**Fig 7** and **S2 Table**).

## Discussion

The problems of sparseness labeling identified in previous studies included the low regulatability of the sparseness level of the effector with low reproducibility. In this study, we developed a novel sparseness labeling method that overcame the weakness of the previous studies. We

**(A)**

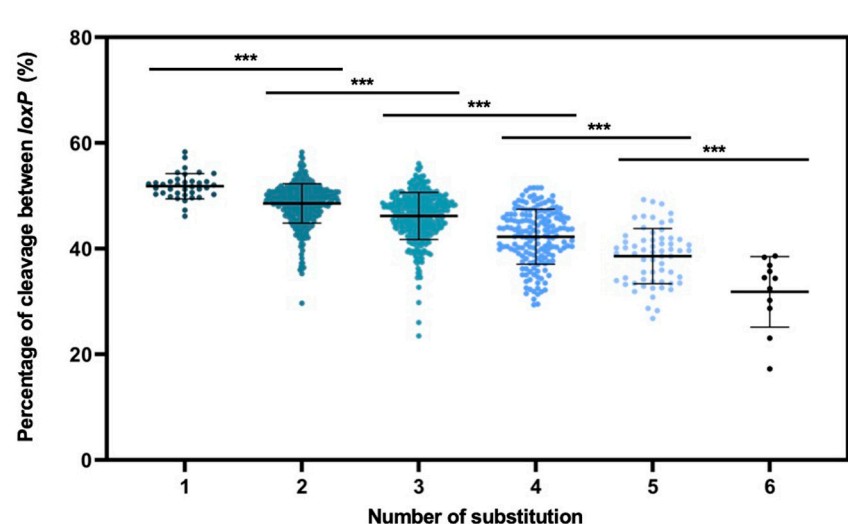

**(B)**

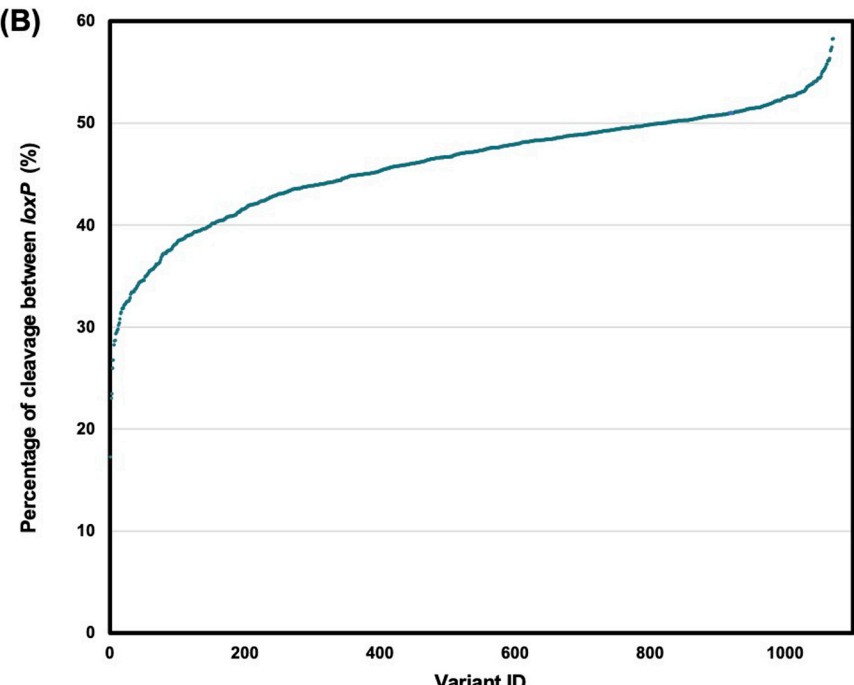

**Fig 5. Illumina sequencing data analysis of the *loxP* variants. (A)** Cleavage rate of all *loxP* variants is classified according to the number of base substitutions. A significant difference was found between each number of substitutions. *** $P < 0.001$, two-tailed *t*-test. **(B)** Distribution of all *loxP* variants. The vertical axis indicates the cleavage rate between *loxP* sequences. *loxP* variants with reduced recognition rate by Cre were listed in order of the cleavage rate between *loxP* sequences. There was some overlap between the plots.

aimed to obtain mutant *loxP* sequences by introducing random mutations into the right RBE sequence according to our hypothesis that *loxP* variants with reduced recognition efficiency by Cre can regulate the sparseness level desirably. We analyzed 1111 variants and successfully identified variants with a recognition efficiency ranging from 17% to 59% (**Fig 5B**).

**Table 1. List of top 50 mutant *loxP* sequences with the lowest recognition rate by Cre among evaluated *loxP* variants.** The underlined bases indicate mutated positions.

| Sequence of *loxP* variants | | | Number of substitutions | Cleavage rate between *loxP* sequences (%) |
|---|---|---|---|---|
| **Left REB** | **Spacer** | **Right RBE** | | |
| ATAACTTCGTATA | -GCATACAT- | GATGTCAAGATAG | 6 | 17.3 |
| ATAACTTCGTATA | -GCATACAT- | TGGAGCATGTCAT | 6 | 23 |
| ATAACTTCGTATA | -GCATACAT- | CGTACAAAGTTAT | 3 | 23.5 |
| ATAACTTCGTATA | -GCATACAT- | TCAACCAAGTTAT | 3 | 26 |
| ATAACTTCGTATA | -GCATACAT- | TGCAACAAGTCAT | 5 | 26.8 |
| ATAACTTCGTATA | -GCATACAT- | TCAACACATTTAT | 5 | 28.3 |
| ATAACTTCGTATA | -GCATACAT- | GATACTTATTGAC | 6 | 28.7 |
| ATAACTTCGTATA | -GCATACAT- | CATACCTCGTCAT | 5 | 28.7 |
| ATAACTTCGTATA | -GCATACAT- | AAATCGAAGTCAT | 4 | 29.4 |
| ATAACTTCGTATA | -GCATACAT- | AAAGCGGAGTTAT | 4 | 29.5 |
| ATAACTTCGTATA | -GCATACAT- | AAAACGAAGTTAT | 2 | 29.7 |
| ATAACTTCGTATA | -GCATACAT- | AAAACCAAGTTAT | 3 | 29.8 |
| ATAACTTCGTATA | -GCATACAT- | TTCACCCAGTTTC | 6 | 30.2 |
| ATAACTTCGTATA | -GCATACAT- | TCAACCAAGTAAT | 4 | 30.4 |
| ATAACTTCGTATA | -GCATACAT- | CTGACGGAGTGAT | 5 | 30.8 |
| ATAACTTCGTATA | -GCATACAT- | AAAACGATGTAAT | 4 | 31.3 |
| ATAACTTCGTATA | -GCATACAT- | TCTACATAGATAT | 4 | 31.5 |
| ATAACTTCGTATA | -GCATACAT- | TACATGACGCTAT | 4 | 31.8 |
| ATAACTTCGTATA | -GCATACAT- | TACATCACCTTAT | 5 | 31.9 |
| ATAACTTCGTATA | -GCATACAT- | TACATAAAGTCAT | 4 | 31.9 |
| ATAACTTCGTATA | -GCATACAT- | TACACTGAGTTAG | 4 | 32.2 |
| ATAACTTCGTATA | -GCATACAT- | CATCCCAAACTAT | 5 | 32.2 |
| ATAACTTCGTATA | -GCATACAT- | GATAAGACGTTGT | 4 | 32.3 |
| ATAACTTCGTATA | -GCATACAT- | GATTAGATGCTAC | 6 | 32.4 |
| ATAACTTCGTATA | -GCATACAT- | TGTAGGCAGATAG | 5 | 32.5 |
| ATAACTTCGTATA | -GCATACAT- | GATATGCAGTCAT | 4 | 32.5 |
| ATAACTTCGTATA | -GCATACAT- | TACACAAATTCAT | 4 | 32.5 |
| ATAACTTCGTATA | -GCATACAT- | GATAAAAAATTAC | 5 | 32.6 |
| ATAACTTCGTATA | -GCATACAT- | CATATGACGTTAT | 3 | 32.7 |
| ATAACTTCGTATA | -GCATACAT- | GATATCACGTTCT | 5 | 32.9 |
| ATAACTTCGTATA | -GCATACAT- | GGTACGCACTTCT | 5 | 33.2 |
| ATAACTTCGTATA | -GCATACAT- | CTTAGCCAGTTAT | 5 | 33.4 |
| ATAACTTCGTATA | -GCATACAT- | TTCACGGGATTAT | 5 | 33.4 |
| ATAACTTCGTATA | -GCATACAT- | GACACCAAGATAT | 4 | 33.5 |
| ATAACTTCGTATA | -GCATACAT- | TACACGGAACTAG | 5 | 33.5 |
| ATAACTTCGTATA | -GCATACAT- | TGTACTAAGTTCG | 4 | 33.5 |
| ATAACTTCGTATA | -GCATACAT- | TCTACTATTTTAA | 5 | 33.7 |
| ATAACTTCGTATA | -GCATACAT- | TGACCGACGTTAG | 5 | 33.7 |
| ATAACTTCGTATA | -GCATACAT- | CATACCAAATTCT | 4 | 33.8 |
| ATAACTTCGTATA | -GCATACAT- | TACACGTTATTCT | 5 | 34 |
| ATAACTTCGTATA | -GCATACAT- | CTTACCAAGTTTT | 4 | 34.1 |
| ATAACTTCGTATA | -GCATACAT- | TTTGTCACGTTAT | 5 | 34.2 |
| ATAACTTCGTATA | -GCATACAT- | TTCTGGAAGTTAT | 4 | 34.3 |
| ATAACTTCGTATA | -GCATACAT- | CGAGCGAACCTAT | 6 | 34.4 |
| ATAACTTCGTATA | -GCATACAT- | CATTCAAAAGTAT | 5 | 34.5 |

(*Continued*)

**Table 1.** (Continued)

| Sequence of *loxP* variants | | | Number of substitutions | Cleavage rate between *loxP* sequences (%) |
|---|---|---|---|---|
| Left REB | Spacer | Right RBE | | |
| ATAACTTCGTATA | -GCATACAT- | GATATGAAATTAT | 3 | 34.5 |
| ATAACTTCGTATA | -GCATACAT- | CATATGTTGTCAA | 6 | 34.5 |
| ATAACTTCGTATA | -GCATACAT- | TGTACGGAGTTCT | 3 | 34.5 |
| ATAACTTCGTATA | -GCATACAT- | GATACCAAGTCAT | 3 | 34.6 |
| ATAACTTCGTATA | -GCATACAT- | CATCGGAATTTAA | 5 | 34.6 |

This result supports our hypothesis that the efficiency of recognition of the *loxP* sequence by Cre can be regulated by precisely introducing mutations into the arm of the *lox* sequence. However, the mutated *loxP* sequences maintained a higher recognition efficiency by Cre than initially expected. Even in the case of three or more base substitutions, the recognition efficiency of *loxP* variants remained relatively high. If we adopt sparse labeling in dense tissues, such as the brain, a mutant that can achieve a high sparsity with a labeling rate of 1% or less is needed. As shown in **Fig 5A**, our experiments confirmed that the efficiency of recognition of *loxP* by Cre decreased as the number of substitutions introduced into the RBE of the *loxP* sequence increases. As shown in Fig 7, we evaluated the recognition rate of mutant *loxP* sequences with more than 7-base substitutions, and we obtained several mutant *loxP* sequences with recognition efficiencies of less than 1%. We also found that some mutants with fewer base substitutions may have lower cleavage rates than mutants with more base substitutions, as shown in Fig 7. This result indicates that the evaluation of individual mutant *loxP* sequences is vital to achieving sparse labeling at any desired rates.

As mentioned in the introduction, several previous studies evaluated the effect of mutations on Cre recombinase or *loxP* sequences [32–43]. Hartung, M. & Kisters-Woike, B. evaluated the effect of the mutation in Cre recombinase [41]. However, it is difficult to precisely regulate the sparseness level by introducing mutations into Cre recombinase. On the other hand, our approach can control the sparseness level only by selecting a mutant *loxP* sequence. Also, we can use existing Cre lines. Missirlis, P. I., *et al.* introduced mutations in the spacer region of the *loxP* sequences [34]. The spacer sequence determines the specificity. Hence, we think that many of the mutants would result in a complete loss of recognition by Cre. Thus, introducing mutations into the spacer region is not an appropriate approach to regulate the sparseness level. Sheren J *et al.* examined the effect of mutations on the spacer region and RBE of the *loxP* sequence, respectively. The method of evaluating mutant *loxP* sequences in our experiment is very different from that of Sheren J *et al.* In our study, the *lox2272* sequence competes with the mutant *loxP* sequence. In contrast, Sheren J *et al.* evaluated mutant *loxP* sequences alone. In our experiment, Cre cannot clave the other *lox* sequence if Cre cleaves one *lox* sequence. On the other hand, if the mutant *loxP* sequence is present alone, the Cre can always cleave the *loxP* sequence while Cre is acting. Therefore, the cleavage rate of mutant *loxP* sequences measured by Sheren J *et al.* under competitive conditions with other *lox* sequences such as *lox2272* is unclear. In addition, we have evaluated cleavage rates for over 1000 mutant *loxP* sequences in this study, allowing us to adjust sparse labeling rates very strictly ranging from 0.51%–59%. In summary, this research is the first large dataset for measuring the cleavage rate of mutant *loxP* sequences in competitive conditions with other *lox* sequences (*lox2272* sequence in this study). Thus, our dataset may allow for regulating sparseness levels at the desired rate.

This study provides the proof of concept for establishing a novel method that allows sparse labeling at desired rates. The novel sparse labeling method proposed in this study has the

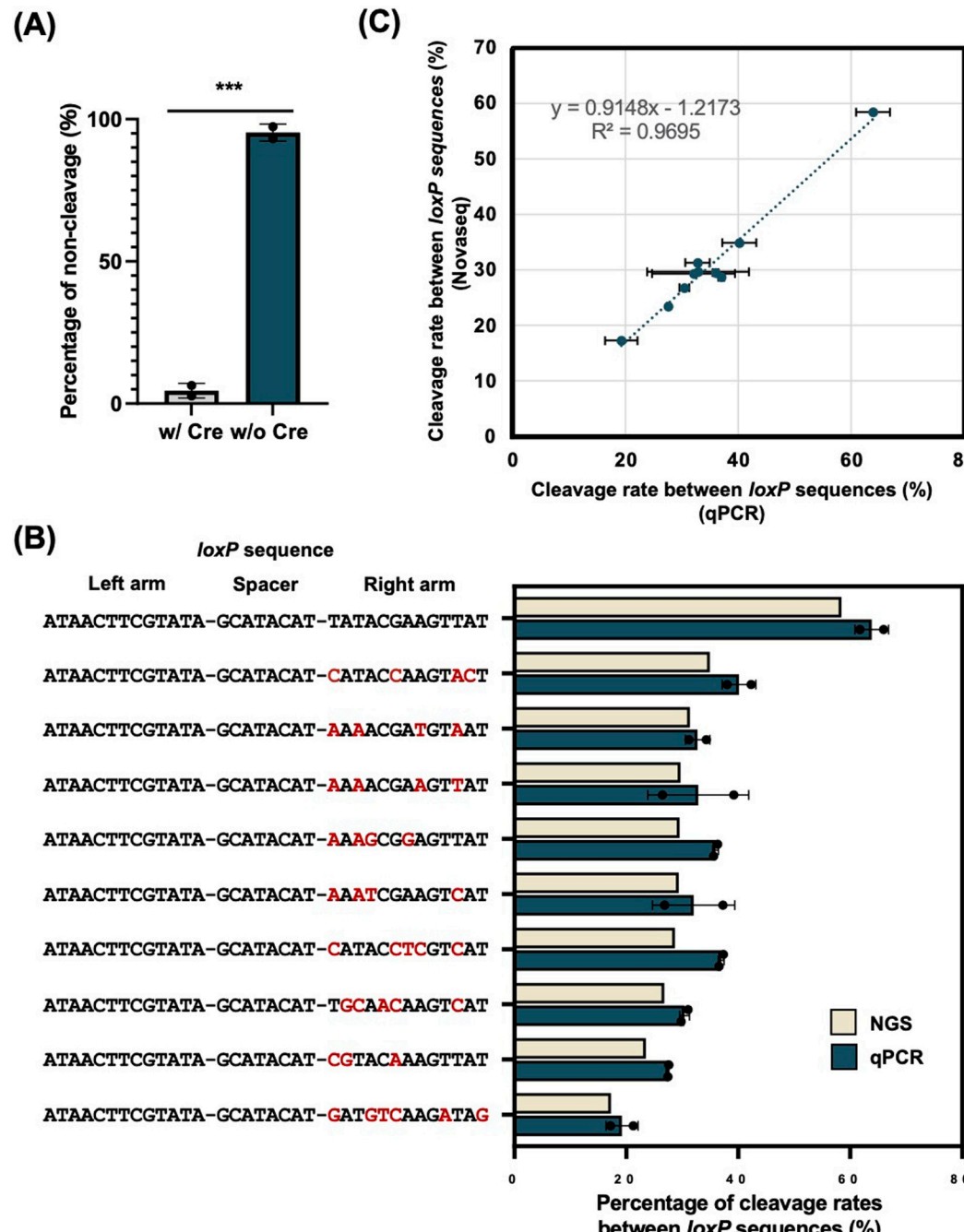

**Fig 6. Quantification of the cleavage rate of *loxP* variants by qPCR. (A)** Comparison of non-cleavage rates in the presence or absence of Cre induction. The error bars represent the standard deviation. A significant difference was found in the presence or absence of Cre induction (two-tailed *t*-test, N = 3). *** $P < 0.001$ **(B)** Comparison of the results of the Illumina sequencing analysis and qPCR. We randomly selected nine *loxP* variants and the WT *loxP* sequence for qPCR. We quantified the cleavage rate of the *loxP* sequence in each variant and WT (see details in the Materials and Methods). The mutated nucleotide is shown in red. The bar graph in light yellow shows the cleavage rate of *loxP* sequences quantified by Illumina sequencing, whereas green indicates the results of qPCR. Each plot was averaged across two independent experiments. The error bars represent the standard deviation. **(C)** Correlation of the cleavage rate between *loxP* sequences, as quantified by Illumina sequencing analysis and qPCR. Each plot was averaged across two independent experiments. The error bars represent the standard deviation.

**Fig 7. The quantification of *loxP* variants with more than 7-base substitutions by qPCR.** We randomly selected *loxP* mutants with more than 7-base substitutions. We quantified the cleavage rate of the each mutant *loxP* sequence using qPCR. The mutated nucleotide is shown in red. Mutants marked with an asterisk showed less than 1% of recognition rate by Cre. The bar graph in green indicates the results of qPCR. Each plot was averaged across two independent experiments. The error bars represent the standard deviation.

potential to overcome some of the disadvantages of previous methods. The screening of transgenic lines with desired expression patterns [5–8], titration of a suitable tamoxifen dosage [9–15], or amount of viral injection [16, 17] require very sophisticated experimental techniques or the time-consuming titration of chemical or gene induction conditions to limit the spatial and temporal expression of a recombinase. In our method, screening or titrating the chemical and genetic induction conditions is unnecessary to achieve the desired sparseness because a mutant *loxP* sequence that can achieve a desired labeling rate is selected in advance. The MORF and MADM/MARCM methods have a disadvantage that the sparseness level cannot be controlled [18–22]. The SPARC and STARS methods have the problem of low controllable sparseness levels [23, 24]. On the other hand, our method can regulate the sparseness level in more than 1000 patterns ranging from 0.51% to 59% by appropriate mutant *loxP* sequence from the mutant *loxP* library.

The ability to regulate the sparseness level at the desired rate with high reproducibility is a major advantage of this method. One example is the combination of our method with the method that uses a cellomics approach [48]. The cellomics approach method applies the Cre-*lox* system to stochastically label opsin in a small population of neural networks. If the mutant *loxP* sequences can be used in the cellomics approach, we could intervene in the activity of a smaller number of neurons. In other examples, our method can be applied to generate a genetic mosaic for analyzing the population dosage of genes involved in sporadic genetic illnesses, as well as to promote cancer development. By applying this method to stepwise change

normal cells into cells expressing cancer-inducing genes in a cell population at desired levels, it may be possible to mimic the environment of cell competition when a minority population of cancer cells is present in the majority of the normal cell population [50–55]. Taken together, as a purely genetic method, it has the potential to be adapted to a variety of fields of research.

Finally, we discuss the limitation of our sparse labeling method. The primary purpose of this study is to present data on the cleavage rate of mutant *loxP* sequencing, which is a proof-of-concept for establishing a novel methodology that allows regulating sparseness levels at the desired rate. We have not yet tested our methodology in neural tissues, and we also have not yet tested the labeling rate in different cell types. As a future experimental plan, it is necessary to test our method in several types of neural tissue and validate the possibility of regulating sparseness levels at the desired rates. Another potential limitation also exists. When combining the mutant *loxP* sequences with the Brainbow system, we can use existing mouse lines for the Cre expression system. For the Brainbow transgenic lines, we have to establish a new line for each mutant *loxP* sequence. There is currently no way to reduce this effort, and it will need to be solved in the future.

## Materials and methods

### Construction of a library of mutant *loxP* sequences

A library of mutant *loxP* sequences was constructed via PCR using the pRS416-*lox* plasmid as a template (S2 File). We mutagenized the *loxP* sequences using primer_No. 1 and primer_No. 2, as shown in S3 Table. The primers were synthesized commercially via solid-phase synthesis (Eurofin, Tokyo, Japan). The underlined 13-base sequence in primer_No. 2 indicates the mutagenized location of the *loxP* sequence. The underlined bases were synthesized using biased randomization with an 84.7% chance of retaining the original sequence (e.g., A means A: 84.7%, T: 5.1%, G: 5.1%, C: 5.1%). This is because this study aimed to evaluate *loxP* mutants with one or two nucleotide substitutions and a certain extent of mutants with three or more nucleotide substitutions. When $x$ = 84.7, the two nucleotide substitutions are most efficiently obtained. We calculated the distribution of the number of base substitutions using the following formula:

$$\frac{13!}{(y!(13-y)!)} \times \left(\frac{x}{100}\right)^{13-y} \times \left(\frac{100-x}{100}\right)^{y}$$

In this formula, the retention rate [%] is represented by $x$ and the number of base substitutions is represented by $y$.

The library was transformed in competent *E. coli* DH5α (F⁻, Φ80d*lacZΔ*M15, *Δ*(*lacZYA-argF*)U169, *deoR*, *recA*1, *endA*1, *hsdR*17(r_K⁻, m_K⁺), *phoA*, *supE*44, λ⁻, *thi*-1, *gyrA*96, *relA*1). The transformed *E. coli* was cultured in a Luria-Bertani (LB) medium (1% [w/v] tryptone, 0.5% [w/v] yeast extract and 1% [w/v] sodium chloride) containing 100 μg/mL ampicillin. Then, plasmids were extracted using the FastGene Plasmid Minikit (NIPPON Genetics, Tokyo, Japan, FG-90502).

### Cre induction in yeast

We constructed β-estradiol-inducible Cre-expressing yeast. The *S. cerevisiae* strain BY4741 (*MATa*, *his3Δ1*, *leu2Δ0*, *met15Δ0*, *ura3Δ0*) was used as the host. The pRS403-CreEBD plasmid (S2 File) was inserted genomically into the yeast *his3Δ1* site. Using a Frozen EZ Yeast Transformation II Kit (Zymo Research, Irvine, CA, USA, T2001), yeast cells were transformed with the constructed plasmid. The transformants were screened on a synthetic defined (SD) solid

medium without L-histidine. The components of the solid SD medium were 0.67% [w/v] yeast nitrogen base without amino acids, 2% [w/v] glucose, and 2% [w/v] agar with appropriate amino acids and a nucleobase (0.012% [w/v] L-leucine, 0.002% [w/v] L-methionine, and 0.002% [w/v] uracil). The mutagenized *loxP* library was transformed into a yeast strain possessing the CreEBD transgene. The transformants were screened on an SD solid medium without L-histidine and uracil. A total of more than 10,000 colonies were collected to prevent the loss of diversity in the *loxP* library. For 24 h, the colonies were precultured on liquid SD medium at 30˚C and 250 rpm. The precultured yeast was inoculated into 10 mL of liquid SD medium containing D-galactose as a carbon source with an $OD_{600}$ of 1. Yeast cells were cultured at 30˚C and 250 rpm for 24 h.

## Preparation of DNA samples for Illumina sequencing analysis

The *loxP* library after Cre induction was extracted using the Zymoprep™ Yeast Plasmid Miniprep II kit (ZYMO RESEARCH, D2004). The extracted samples were amplified by PCR using primer_No. 3 and primer_No. 4. The cycling parameters were as follows: 94˚C for 2 min; followed by five cycles at 98˚C for 10 s/55˚C for 5 s/68˚C for 30 s, 68˚C for 7 min; and final hold at 4˚C. The number of PCR cycles was set to five to reduce PCR bias as much as possible. The prepared samples were sequenced using Novaseq6000 (Illumina, San Diego, CA, USA) at the paired end. The quality check of samples, the addition of adaptor sequences, the addition of index sequences, and Illumina sequencing runs were performed using the services of Macro-Gen Japan (Tokyo, Japan).

## Illumina sequencing data analysis

The Python script shown in **S3 File** was used to analyze the Illumina sequencing data. Briefly, using fastq files and the Python script, the sequences of *loxP* variants were extracted and the number of reads was counted. We set the cut-off for analysis at 500 to obtain reliable *loxP* cleavage rates. We classified the cleavage patterns of each Illumina sequencing read (noncleavage, cleavage between *loxP*, and cleavage between *lox2272*) using the paired-end sequence information. We calculated the cleavage rate between *loxP* sequences based on the following formula:

$$\frac{\text{The number of NGS reads cleaved between } loxP}{\text{The number of NGS reads cleaved between } loxP \text{ or } lox2272} \times 100$$

The processed data are presented in **S1 File**.

## Quantitative polymerase chain reaction (qPCR)

To confirm the accuracy of the results of the Illumina sequencing analysis, we conducted a validation experiment by quantitative polymerase chain reaction (qPCR). Nine *loxP* variants were selected randomly from sequences with a decreased cleavage rate compared with WT *loxP*. We constructed plasmids that possessed the target mutation via PCR using pRS416_Leu_Ura (**S2 File**) as a template. The primers used to amplify the DNA fragments are shown in **S3 Table** (primer_No. 5 to No. 15). Then, we prepared three sets of primers for qPCR: (1) primers to quantify the cleavage rate between *lox2272* (primer_No. 16 and primer_No. 17), (2) primers to quantify the cleavage rate between *loxP* (primer_No. 18 and primer_No. 19), and (3) primers to quantify the non-cleavage rate (primer_No. 20 and primer_No. 21). In addition, we used pRS416_Leu_Ura, pRS416_Leu_Ura_ΔloxP, and pRS416_Leu_Ura_Δlox2272 (**S2 File**) for generating a calibration curve. A dilution series of 7 points in 10-fold increments from $1.0 \times 10^{7}$ copies/μL to 1 copy/μL was used for the calibration

curve. The non-cleavage rate, the cleavage rate between *loxP*, and the cleavage rate between *lox2272* of individual mutant *loxP* sequences were calculated using these calibration curves. The PCR mixture included: 17 μL of distilled water, 25 μL of FastStart SYBR Green Master (without ROX) (Roche, Basel, Switzerland, 04673484001), 1.5 μL of 10 pmol/μL forward primer, 1.5 μL of 10 pmol/μL reverse primer, and 5 μL of the template plasmid. PCR was carried out on a StepOnePlus™ instrument (Thermo Fisher Scientific, USA) using the following cycling conditions for all primer sets: 95˚C for 10 min; followed by 40 cycles of 95˚C for 15s, 60˚C for 30s; and 1 cycle of 95˚C for 15s, 60˚C for 1 min, and 95˚C for 10 s.

## Supporting information

**S1 Table. Sequences of *loxP* variants confirmed by Sanger sequencing.**
(XLSX)

**S2 Table. List of *loxP* variants measured by qPCR.**
(XLSX)

**S3 Table. Primers used in this study.**
(XLSX)

**S1 File. List of all *loxP* variants evaluated in this research.**
(XLSX)

**S2 File. Plasmid maps used in this study.** The full sequences of the plasmids used in this study are shown.
(XLSX)

**S3 File. Python scripts for data analysis.**
(TXT)

## Author Contributions

**Conceptualization:** Yuji Yamauchi, Wataru Aoki.

**Data curation:** Yuji Yamauchi, Hidenori Matsukura, Wataru Aoki.

**Formal analysis:** Yuji Yamauchi.

**Funding acquisition:** Yuji Yamauchi, Wataru Aoki.

**Investigation:** Yuji Yamauchi, Hidenori Matsukura.

**Methodology:** Yuji Yamauchi, Hidenori Matsukura, Wataru Aoki.

**Project administration:** Yuji Yamauchi, Wataru Aoki.

**Resources:** Yuji Yamauchi, Hidenori Matsukura.

**Software:** Yuji Yamauchi, Keisuke Motone.

**Supervision:** Mitsuyoshi Ueda, Wataru Aoki.

**Validation:** Yuji Yamauchi, Hidenori Matsukura.

**Visualization:** Yuji Yamauchi.

**Writing – original draft:** Yuji Yamauchi, Wataru Aoki.

**Writing – review & editing:** Yuji Yamauchi, Hidenori Matsukura, Keisuke Motone, Mitsuyoshi Ueda, Wataru Aoki.

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
