## [Decision Letter · Decision Letter 0]

4 Aug 2022

PONE-D-22-18681Evaluation of a library of loxP variants for a novel sparse labeling strategyPLOS ONE

Dear Dr. Aoki,

Thank you for submitting your manuscript to PLOS ONE. After careful consideration, we feel that it has merit but does not fully meet PLOS ONE’s publication criteria as it currently stands. Therefore, we invite you to submit a revised version of the manuscript that addresses the points raised during the review process.

A library of mutant loxP sequences with differing recombination efficiency could be a valuable resource. However, all the reviewers raised the question that the low recombination efficiency of the loxp mutants may not achieve the aim of sparse labeling of mammalian cells/neurons. Please provide the extended experimental evidence or extensively revise the manuscript to tune down the claim and address the technical issue (e.g., PCR bias) suggested by the reviewers.

We look forward to receiving your revised manuscript.

Kind regards,

Xiao-Hong Lu, Ph.D.

Academic Editor

PLOS ONE

Journal Requirements:

Reviewers' comments:

Reviewer's Responses to Questions

**Comments to the Author**

1. Is the manuscript technically sound, and do the data support the conclusions?

Reviewer #1: Partly

Reviewer #2: Partly

Reviewer #3: Partly

2. Has the statistical analysis been performed appropriately and rigorously? 

Reviewer #1: Yes

Reviewer #2: N/A

Reviewer #3: Yes

3. Have the authors made all data underlying the findings in their manuscript fully available?

Reviewer #1: Yes

Reviewer #2: Yes

Reviewer #3: Yes

4. Is the manuscript presented in an intelligible fashion and written in standard English?

Reviewer #1: Yes

Reviewer #2: Yes

Reviewer #3: Yes

5. Review Comments to the Author

Reviewer #1: The manuscript describes the generation and characterization of a library of mutant loxP sequences with differing recombination efficiency. The approach used is appropriate and the experiments are described well and the data are presented clearly.

The major limitation of the study is that while the stated goal of the study is to generate a library of LoxP mutants that will facilitate spare labelling, the library generated has recognition efficiencies ranging from 17 % to 59 %. As the authors acknowledge, efficiencies in the low single digit range (1 or 2%) would be required for spare labelling of cells. Thus the utility of the library is somewhat overstated.

To achieve the goal of achieving low recognition efficiencies the authors should be able to use the data already generated to identify positions within the RBE at which substitutions have the greatest effect on recognition efficiency. This information could then be used to guide the selection of combinations of substitutions that would achieve low recognition efficiencies. Extending the work in this way would greatly enhance its value and potential impact and application.

It would be interesting to know what would be the effect of completely eliminating or randomizing one RBE. Does this completely eliminate recognition by cre? Perhaps this information is available in the literature, but if not it could be determined experimentally. This would establish the maximum decrease in efficiency that can be achieved with the chosen approach.

Typographical errors

Line 137 Design of a lirary of mutant loxP sequences. CHANGE Lirary to library.

Reviewer #2: This study from Yamauchi, et al., details the process of generating a library of mutant loxP sequences as a tool for precisely modulating Cre binding and genetic labeling at desired levels of sparseness. This highly focused study generated, characterized, and validated the decrease in Cre-mediated cleavage resulting from random mutagenesis of the loxP recombinase binding element (RBE). The generated loxP sequence cleavage rate data may support increasingly precise biological manipulation by scientists in a variety of fields. Below are some comments regarding the authors’ presentation of their methodologies and data:

Major issue:

The authors present their novel loxP-based sparse labeling strategy as an improvement over the existing sparse labeling techniques due to its ability to precisely modulate the labeling density. As with Brainbow, the need for sparse labeling in the visualization of neurons is given as a source of demand for improved labeling technologies. However, the authors did not test their methodology in neurons. Additionally, the variable labeling rate in different cell types was given as a shortcoming of other techniques, but the authors did not test their strategy in different cell types. Finally, visualization of neurons requires a labeling sparseness level of ~1%. The reduced cleavage rate of this paper’s modified loxP sequences was not below 25%. If the authors believe that their system would result in a ~1% labeling rate in neurons in vitro or in vivo, this should be either demonstrated or extensively discussed. Ultimately, the authors should address whether and how their novel labeling strategy addresses the shortcomings given for existing labeling methods. If performing the experiments necessary to fully characterize this system in neurons and/or multiple cell types would be prohibitive, then a significant portion of the Discussion section should be dedicated to this matter.

Minor issues / comments:

Abstract: As discussed above, while the methodologies detailed in this research are based upon the concepts behind the Brainbow system, the author’s novel labeling system was not “adopted […] to stochastically label cells” in this paper. Rather, this article focuses on purely genetic analysis of Cre-based cleavage efficiency, which should be emphasized in order to generate accurate expectations from the reader.

Introduction: This introduction details both the need for tools to stochastically and sparsely label cell populations, as well as the shortcomings of several existing systems. Grammatical proofreading of this section is advised.

Results and Figures/Figure Legends: This section is appropriately detailed and clearly walks the reader through the generation, analysis, and validation of loxP mutations. A few minor points should be addressed:

Figure 1B: Less cleavage at loxP results in less excision / relatively more expression of gene B, as indicated in Figure 2D. I believe that Figure 1B incorrectly conveys that the library of loxP variants (with lower Cre binding affinity) will produce lower levels of gene B expression.

Figure 2 legend: Cre induction is mentioned repeatedly in the text but not shown on the figure; may want to indicate β-estradiol treatment.

Figure 3 legend: Please clarify how you “set the mutation rate of the primer to 15.3%”.

Figure 4A(3) is not referenced in the text, while all other subsections of the figure are referenced.

It is stated that an average non-cleavage rate of 3.5% confirms that Cre is correctly induced – what is the acceptable range of non-cleavage values that would indicate Cre induction?

A brief explanation of the advantages of qPCR analysis over Illumina sequencing should be provided to support qPCR as a necessary and appropriate method of validation.

Proofreading of this section as well as supplementary materials is advised (typographical errors present).

Discussion: Good, concise summary detailing the outcomes of the study and potential future applications in a variety of fields.

Materials and Methods: In referencing the primers used to amplify the DNA fragments for qPCR from S2 Table, primer_No. 5 is never referenced. Additionally, primer_No. 21 is listed as a primer to quantify the non-cleavage rate, while S2 Table contains no primer_No. 21. I believe there has been a transposition error regarding the numbering of Table S2. (Of note, formatting of primers on S2 Table is not consistent.)

Reviewer #3: In this study, the authors evaluated a library of loxP variants for Cre-mediated excision efficiency using sequencing in yeast. Despite the efforts of the authors, the values of this study to the field of science is limited.

Since cre-lox site-specific recombination system first developed in the yeast in 1987, mutations on loxP and the Cre recombinase have been extensively examined. Mutagenic studies of loxP have shown that many mutations, in either the 8 bp spacer region or the two 13 bp inverted repeats, affect the recombination efficiency. Here are a couple examples: Hartung, M. & Kisters-Woike, B. Cre mutants with altered DNA binding properties. J. Biol. Chem. 273, 22884–22891 (1998); Missirlis, P. I., Smailus, D. E. & Holt, R. A. A high-throughput screen identifying sequence and promiscuity characteristics of the loxP spacer region in Cre-mediated recombination. BMC Genomics 7, 73 (2006); Sheren, J., Langer, S. J. & Leinwand, L. A. A randomized library approach to identifying functional lox site domains for the Cre recombinase. Nucleic Acids Res. 35, 5464–5473 (2007).

Over a dozen different loxP variants have been stringently tested in mammalian cells and higher systems, so it is a validated fact, not a hypothesis as the authors claimed “that the efficiency of recognition of the loxP sequence by Cre can be regulated by introducing mutations into the arm of the lox sequence”.

The authors used many paragraphs to describe sparse labeling and Brainbow system, but their study has nothing to do with real “labeling”, even in mammalian cells. So “a novel sparse labeling strategy” is vastly overclaimed by the authors. Using low efficiency loxP sites for sparse labeling may be a reasonable idea, but many candidate variants have been identified from previous studies, such as above mentioned Sheren et al. 2007 study listed at least dozens of loxP variants with less than 5% recombination rate, which is much lower than the lowest percentage of cleavage the authors got in this study (~20%).

Lastly, Illumina sequencing may not be the best method for the measurement of recombination efficiency, even with 5 cycles, the PCR bias may affect the accuracy. Nanostring or other amplification-free technology that can count the DNA molecules directly are better options.

6. PLOS authors have the option to publish the peer review history of their article (what does this mean?). If published, this will include your full peer review and any attached files.

Reviewer #1: No

Reviewer #2: **Yes: **Erika Knott Reece

Reviewer #3: No

---

## [Author Response · Author response to Decision Letter 0]

17 Sep 2022

Response to Reviewer #1’s comments

Reviewer #1’s comment 1:

The manuscript describes the generation and characterization of a library of mutant loxP sequences with differing recombination efficiency. The approach used is appropriate and the experiments are described well and the data are presented clearly.

Authors’ response 1:

First of all, we thank the reviewer #1 for improving our original manuscript. We also appreciate the time and effort that you have dedicated to provide insightful feedbacks. We are glad that you have appreciated our paper.

Reviewer #1’s comment 2:

The major limitation of the study is that while the stated goal of the study is to generate a library of loxP mutants that will facilitate spare labelling, the library generated has recognition efficiencies ranging from 17 % to 59 %. As the authors acknowledge, efficiencies in the low single digit range (1 or 2%) would be required for spare labelling of cells. Thus the utility of the library is somewhat overstated.

Authors’ response 2:

As you mentioned, we did not acquire mutant loxP sequence with efficiencies in the low single-digit range (1 or 2%). Fig. 5A indicated that the greater the base substitution number, the lower the recognition rate by Cre; hence we hypothesized that by evaluating mutant loxP sequences with complex mutation patterns, we could obtain mutant loxP sequences with lower cleavage efficiencies. To obtain mutant loxP sequences with lower recognition efficiency by Cre, we evaluated the recognition rate of mutant loxP sequences with more than 7-base substitutions. We successfully acquired mutant loxP sequences with less than 1 % cleavage rate in this revision. We added a new Figure (Fig. 7) and revised the manuscripts (Fig. 7, lines 249–264, 282–287 in the revised manuscript). 

Reviewer #1’s comment 3:

To achieve the goal of achieving low recognition efficiencies the authors should be able to use the data already generated to identify positions within the RBE at which substitutions have the greatest effect on recognition efficiency. This information could then be used to guide the selection of combinations of substitutions that would achieve low recognition efficiencies. Extending the work in this way would greatly enhance its value and potential impact and application.

Authors’ response 3:

We appreciate your suggestion. We analyzed which positions within RBEs in the loxP sequence were important on cleavage by Cre. The result of the analysis suggested that several positions within RBEs are important. However, the library size was somewhat insufficient to draw any conclusions. Therefore, we are planning to expand the library size and identify important positions within the RBEs in further studies.

Reviewer #1’s comment 4:

It would be interesting to know what would be the effect of completely eliminating or randomizing one RBE. Does this completely eliminate recognition by cre? Perhaps this information is available in the literature, but if not it could be determined experimentally. This would establish the maximum decrease in efficiency that can be achieved with the chosen approach.

Authors’ response 4:

We appreciate your suggestion. The effect of completely eliminating or randomizing one RBE in the loxP sequence under competitive conditions with the lox2272 sequence has not been investigated. We measured a mutant loxP sequence with a completely randomized RBE (Fig. 7; Right arm sequence, GGCCAACGAAGCG). As a result, this mutant showed less than 1% cleavage efficiency by Cre. We revised the manuscripts (Fig. 7, lines 249–264, 282–287 in the revised manuscript). 

Reviewer #1’s comment 5

Typographical errors

Line 137 Design of a lirary of mutant loxP sequences. CHANGE Lirary to library.

Authors’ response 5:

Thank you for pointing this out. We have corrected the relevant section in the manuscript from lirary to library (line 137 in the revised manuscript).

Response to Reviewer #2’s comments

Reviewer #2’s comment 1:

This study from Yamauchi, et al., details the process of generating a library of mutant loxP sequences as a tool for precisely modulating Cre binding and genetic labeling at desired levels of sparseness. This highly focused study generated, characterized, and validated the decrease in Cre-mediated cleavage resulting from random mutagenesis of the loxP recombinase binding element (RBE). The generated loxP sequence cleavage rate data may support increasingly precise biological manipulation by scientists in a variety of fields. Below are some comments regarding the authors’ presentation of their methodologies and data:

Authors’ response 1:

First of all, we thank the reviewer #2 for improving our original manuscript. We appreciate the time and effort that you have dedicated to provide insightful feedbacks. We are glad that you have appreciated our paper such as “This highly focused study”and “The generated loxP sequence cleavage rate data may support increasingly precise biological manipulation by scientists in a variety of fields.”

Reviewer #2’s comment 2:

The authors present their novel loxP-based sparse labeling strategy as an improvement over the existing sparse labeling techniques due to its ability to precisely modulate the labeling density. As with Brainbow, the need for sparse labeling in the visualization of neurons is given as a source of demand for improved labeling technologies. However, the authors did not test their methodology in neurons. Additionally, the variable labeling rate in different cell types was given as a shortcoming of other techniques, but the authors did not test their strategy in different cell types.

Authors’ response 2:

As you pointed out, investigating our methodology in neurons and several different cell types is crucial. However, the primary purpose of this paper is to propose the proof of concept of a new sparse labeling method. In the future study, we will test our strategy in neurons and different types of cells to propose the utility of our method. We modified our manuscript to clarify that the primary purpose of this study is to propose a new sparse labeling method and that there is a need to perform sparse labeling on multiple types of neurons in the future (lines 84–86, lines 335–341 in the revised manuscripts).

Reviewer #2’s comment 3:

Finally, visualization of neurons requires a labeling sparseness level of ~1%. The reduced cleavage rate of this paper’s modified loxP sequences was not below 25%. If the authors believe that their system would result in a ~1% labeling rate in neurons in vitro or in vivo, this should be either demonstrated or extensively discussed.

Authors’ response 3:

As you pointed out, we could not acquire mutant loxP sequence with less than 1 % cleavage rate by Cre in the original manuscript. Thus, we acquired additional mutant loxP sequences with less than 1 % cleavage rate in this revision. We added a new Figure (Fig. 7) and revised the manuscript (Fig. 7, lines 249–264, 282–287 in the revised manuscript).

Reviewer #2’s comment 4:

Ultimately, the authors should address whether and how their novel labeling strategy addresses the shortcomings given for existing labeling methods. If performing the experiments necessary to fully characterize this system in neurons and/or multiple cell types would be prohibitive, then a significant portion of the Discussion section should be dedicated to this matter.

Authors’ response 4:

We agree with you. The purpose of this paper was to demonstrate the proof-of-concept of a new sparse labeling method. The most significant advantage of this methodology is that the sparseness level can be controlled (up to 1000 patterns or more) using the mutant loxP sequences obtained in this study. This advantage is not found in existing sparse labeling methods. We explained how our method could solve the disadvantages of each existing method in the discussion (lines 310–322 in the revised manuscript). In addition, since this study did not demonstrate the utility of our method in neurons, we discuss it as a limitation of this study (lines 335–341 in the revised manuscript).

Reviewer #2’s comment 5:

Minor issues / comments:

Abstract: As discussed above, while the methodologies detailed in this research are based upon the concepts behind the Brainbow system, the author’s novel labeling system was not “adopted […] to stochastically label cells” in this paper. Rather, this article focuses on purely genetic analysis of Cre-based cleavage efficiency, which should be emphasized in order to generate accurate expectations from the reader.

Authors’ response 5:

We agree with you. We edited Abstract according to your comment (lines 22–23 in the revised manuscript).

Reviewer #2’s comment 6: 

Introduction: This introduction details both the need for tools to stochastically and sparsely label cell populations, as well as the shortcomings of several existing systems. Grammatical proofreading of this section is advised.

Authors’ response 6:

Thank you for your suggestion. Grammar proofreading was conducted by Enago.

Reviewer #2’s comment 7:

Results and Figures/Figure Legends: 

This section is appropriately detailed and clearly walks the reader through the generation, analysis, and validation of loxP mutations. A few minor points should be addressed:

Figure 1B: Less cleavage at loxP results in less excision / relatively more expression of gene B, as indicated in Figure 2D. I believe that Figure 1B incorrectly conveys that the library of loxP variants (with lower Cre binding affinity) will produce lower levels of gene B expression.

Authors’ response 7:

We agree with you. The library of loxP variants (with lower Cre binding affinity) will produce lower levels of gene A expression. We revised Fig. 1B according to your suggestion.

Reviewer #2’s comment 8:

Figure 2 legend: Cre induction is mentioned repeatedly in the text but not shown on the figure; may want to indicate β-estradiol treatment.

Authors’ response 8:

We agree with you. We modified Figure 2 to clearly convey that Cre induction was performed by β-estradiol.

Reviewer #2’s comment 9:

Figure 3 legend: Please clarify how you “set the mutation rate of the primer to 15.3%”.

Authors’ response 9:

Due to the limitation of the number of reads in Illumina sequencing, the goal of this study was to analyze up to 2-base substitutions. When the mutated rate in randomized primer is 15.3 %, 2-base substitutions are most efficiently obtained. We described the reason why we set the mutation rate of the rimer to 15.3 % (lines 148–149, 159–160 in the revised manuscript).

Reviewer #2’s comment 10:

Figure 4A(3) is not referenced in the text, while all other subsections of the figure are referenced.

Authors’ response 10:

Thank you for pointing this out. We referred Fig. 4A (3) and corrected the relevant section in the manuscript (line 174 in the revised manuscript).

Reviewer #2’s comment 11:

It is stated that an average non-cleavage rate of 3.5% confirms that Cre is correctly induced – what is the acceptable range of non-cleavage values that would indicate Cre induction?

Authors’ response 11:

Thank you for providing important insight. In Figure 6A, we quantified the non-cleavage rate in the absence or presence of Cre induction by qPCR. A non-cleavage rate in the absence of Cre induction was over 90 %. Based on this result, we consider that if the non-cleavage rate is less than 10 %, the induction of Cre is successfully conducted.

Reviewer #2’s comment 12:

A brief explanation of the advantages of qPCR analysis over Illumina sequencing should be provided to support qPCR as a necessary and appropriate method of validation.

Authors’ response 12:

We agree with you. While Illumina sequencing is high-throughput, the results of Illumina sequencing potentially have a certain bias because it requires PCR during sample preparation. If the bias is significant, we cannot trust the cleavage rates of the mutant loxP sequences measured by Illumina sequencing for sparse labeling. To confirm that there is no significant bias in the results of Illumina sequencing, we used qPCR, which has the disadvantage of low- throughput but can accurately quantify the cleavage rates by Cre. We edited the manuscript to clarify the necessity of the validation experiment and the reason for choosing qPCR (lines 222–226 in the revised manuscript).

Reviewer #2’s comment 13:

Proofreading of this section as well as supplementary materials is advised (typographical errors present).

Authors’ response 13:

Thank you for your suggestion. Grammar proofreading was conducted by Enago.

Reviewer #2’s comment 14:

Discussion: Good, concise summary detailing the outcomes of the study and potential future applications in a variety of fields.

Authors’ response 14:

Thank you for your comment. We are glad about your evaluation.

Reviewer #2’s comment 15:

Materials and Methods: In referencing the primers used to amplify the DNA fragments for qPCR from S2 Table, primer_No. 5 is never referenced. Additionally, primer_No. 21 is listed as a primer to quantify the non-cleavage rate, while S2 Table contains no primer_No. 21. I believe there has been a transposition error regarding the numbering of Table S2. (Of note, formatting of primers on S2 Table is not consistent.)

Authors’ response 15:

We agree with you. We referred to primer_No. 5 in the manuscript (lines 414 in the revised manuscripts). We also listed primer_No. 21 in S2 Table. All primers for S2 Table were standardized with capital letters.

 

Response to Reviewer #3’s comments

Reviewer #3’s comment 1:

In this study, the authors evaluated a library of loxP variants for Cre-mediated excision efficiency using sequencing in yeast. Despite the efforts of the authors, the values of this study to the field of science is limited. Since cre-lox site-specific recombination system first developed in the yeast in 1987, mutations on loxP and the Cre recombinase have been extensively examined. Mutagenic studies of loxP have shown that many mutations, in either the 8 bp spacer region or the two 13 bp inverted repeats, affect the recombination efficiency. Here are a couple examples: Hartung, M. & Kisters-Woike, B. Cre mutants with altered DNA binding properties. J. Biol. Chem. 273, 22884–22891 (1998); Missirlis, P. I., Smailus, D. E. & Holt, R. A. A high-throughput screen identifying sequence and promiscuity characteristics of the loxP spacer region in Cre-mediated recombination. BMC Genomics 7, 73 (2006); Sheren, J., Langer, S. J. & Leinwand, L. A. A randomized library approach to identifying functional lox site domains for the Cre recombinase. Nucleic Acids Res. 35, 5464–5473 (2007).

Authors’ response 1:

First of all, we thank the reviewer #3 for improving our original manuscript. We appreciate the time and effort that you have dedicated to provide insightful feedbacks. 

As you mentioned, several previous studies evaluated the effect of mutations on Cre recombinase or loxP sequences. The purpose of this paper was to demonstrate the proof-of-concept of a new sparse labeling method. The most significant advantage of this methodology is that the sparseness level can be controlled (up to 1000 patterns or more) using mutant loxP sequences obtained in this study. This advantage is not found in existing sparse labeling methods as described in Author’s response 4 for reviewer #2’s comment 4. Also, no other papers have evaluated the effect of mutation on RBE in the loxP sequence under the competitive conditions with the lox2272 sequence. Thus, we consider the results of this study are novel. Hartung, M. & Kisters-Woike, B. evaluated the effect of the mutation in Cre recombinase [41]. However, it is difficult to precisely regulate the sparseness level by introducing mutations into Cre recombinase. On the other hand, our approach can control the sparseness level only by selecting a mutant loxP sequence. Also, we can use existing Cre lines. Missirlis, P. I., et al. introduced mutations in the spacer region of the loxP sequences [34]. The spacer sequence determines the specificity. Hence, we think that many of the mutants would result in a complete loss of recognition by Cre. Thus, introducing mutations into the spacer region is not an appropriate approach to regulate the sparseness level. Sheren J et al. examined the effect of mutations on the spacer region and RBE of the loxP sequence, respectively. The method of evaluating mutant loxP sequences in our experiment is very different from that of Sheren J et al. In our study, the lox2272 sequence competes with the mutant loxP sequence. In contrast, Sheren J et al. evaluated mutant loxP sequences alone. In our experiment, Cre cannot clave the other lox sequence if Cre cleaves one lox sequence. On the other hand, if the mutant loxP sequence is present alone, the Cre can always cleave the loxP sequence while Cre is acting. Therefore, the cleavage rate of mutant loxP sequences measured by Sheren J et al. under competitive conditions with other lox sequences such as lox2272 is unclear. In summary, this research is the first large dataset for measuring the cleavage rate of mutant loxP sequences in competitive conditions with other lox sequences (lox2272 sequence in this study). We added these explanations in the Discussion (lines 288–309 in the revised manuscript).

Reviewer #3’s comment 2:

Over a dozen different loxP variants have been stringently tested in mammalian cells and higher systems, so it is a validated fact, not a hypothesis as the authors claimed “that the efficiency of recognition of the loxP sequence by Cre can be regulated by introducing mutations into the arm of the lox sequence”.

Authors’ response 2:

Thank you for your insightful comment. As we mention in Authors’ response 1 for Reviewer #3’s comment 1, the cleavage between loxP sequences is under the competitive condition with cleavage between lox2272 sequences in our experiment. In addition, we have evaluated cleavage rates for over 1000 mutant loxP sequences in this study, allowing us to adjust sparse labeling rates very strictly ranging from 0.51%–59 %. We revised the Discussion section (line 306–309 in the revised manuscript).

Reviewer #3’s comment 3:

The authors used many paragraphs to describe sparse labeling and Brainbow system, but their study has nothing to do with real “labeling”, even in mammalian cells. So “a novel sparse labeling strategy” is vastly overclaimed by the authors. Using low efficiency loxP sites for sparse labeling may be a reasonable idea, but many candidate variants have been identified from previous studies, such as above mentioned Sheren et al. 2007 study listed at least dozens of loxP variants with less than 5% recombination rate, which is much lower than the lowest percentage of cleavage the authors got in this study (~20%).

Authors’ response 3:

As you pointed out, Sheren et al. obtained at least dozens of loxP variants with less than 5 % recombination rate. However, the sparse labeling method we aimed for in this study is a methodology in which the sparseness level can be regulated at the desired levels. We have measured the cleavage rate of the more than 1000 mutant loxP sequences. The number of measured mutants is more than that of Sheren et al. measured. And we measured the cleavage rate under competitive conditions with other lox sequences, as mentioned above. We consider our results are useful because the conditions under competition with other lox sequences are universally used in Brainbow. The “novel” in the title of our paper, “novel sparse labeling strategy,” is used in the sense of “a novel sparse labeling method that enables multi-step adjustment of sparseness level, which could not be done before.” As you mentioned, we did not investigate our methodology in neurons. This is the limitation of this study. However, the primary purpose of this paper is to propose the proof of concept of a new sparse labeling method. In the future study, we will test our strategy in neurons and different types of cells to propose the utility of our method. We modified the manuscript to clarify the primary purpose of this study and that there is a need to perform sparse labeling on multiple types of neurons in the future (lines 84–86, lines 339–345 in the revised manuscripts). As you mentioned, we did not acquire mutant loxP sequence with efficiencies in the low single-digit range (less than 5 %) in the original manuscripts. Thus, we acquired new mutant loxP sequences with less than 1 % cleavage rate in this revision. We added a new Figure (Fig. 7) and manuscripts (lines 249–264, 282–287 in the revised manuscript).

Reviewer #3’s comment 4:

Lastly, Illumina sequencing may not be the best method for the measurement of recombination efficiency, even with 5 cycles, the PCR bias may affect the accuracy. Nanostring or other amplification-free technology that can count the DNA molecules directly are better options.

Authors’ response 4:

Thank you for your insightful comment. As you mentioned, the results of Illumina sequencing potentially have a certain bias. However, Illumina sequencing has the advantage of high-throughput evaluation, and Illumina sequencing is very cheap. We considered amplicon-free technology but could not acquire the sufficient quality of the sample extracted from yeast under amplicon-free conditions. Of course, we are concerned about PCR bias. Therefore, we limited the PCR cycles to only 5 cycles, and the results obtained by Illumina sequencing were validated by qPCR. The qPCR results show a correlation of R2>0.99 (Figure 5A) compared with the Illumina sequencing results, proving that the data are very high quality. We edited the manuscript to clarify the purpose of qPCR (lines 222–226 in the revised manuscript).

---

## [Decision Letter · Decision Letter 1]

3 Oct 2022

PONE-D-22-18681R1Evaluation of a library of loxP variants for a novel sparse labeling strategyPLOS ONE

Dear Dr. Aoki,

Thank you for submitting your manuscript to PLOS ONE. After careful consideration, we feel that it has merit but does not fully meet PLOS ONE’s publication criteria as it currently stands. Therefore, we invite you to submit a revised version of the manuscript that addresses the points raised during the review process.

We look forward to receiving your revised manuscript.

Kind regards,

Xiao-Hong Lu, M.D., Ph.D.

Academic Editor

PLOS ONE

Journal Requirements:

Reviewers' comments:

Reviewer's Responses to Questions

**Comments to the Author**

1. If the authors have adequately addressed your comments raised in a previous round of review and you feel that this manuscript is now acceptable for publication, you may indicate that here to bypass the “Comments to the Author” section, enter your conflict of interest statement in the “Confidential to Editor” section, and submit your "Accept" recommendation.

Reviewer #1: (No Response)

Reviewer #2: All comments have been addressed

Reviewer #3: All comments have been addressed

2. Is the manuscript technically sound, and do the data support the conclusions?

Reviewer #1: Yes

Reviewer #2: Yes

Reviewer #3: Yes

3. Has the statistical analysis been performed appropriately and rigorously? 

Reviewer #1: Yes

Reviewer #2: N/A

Reviewer #3: Yes

4. Have the authors made all data underlying the findings in their manuscript fully available?

Reviewer #1: Yes

Reviewer #2: Yes

Reviewer #3: Yes

5. Is the manuscript presented in an intelligible fashion and written in standard English?

Reviewer #1: Yes

Reviewer #2: Yes

Reviewer #3: Yes

6. Review Comments to the Author

Reviewer #1: The Authors have addressed the substantial points made in my initial review of the manuscript by including a description of mutants with lower recombination efficiency. I think that the mutants identified represent a valuable resource. However, I think that that the title still overstates what has been done. Mentioning "a novel sparse labelling strategy" is not appropriate when no labelling has been done. Perhaps "Evaluation of a library of loxP variants with a wide range of recombination efficiencies" or something along those lines is

appropriate. Also the title of first results section "Strategy for achieving sparseness labeling at the desired rate" should be changed to "Strategy for achieving recombination efficiency at the desired rate" or something like that?

Also in the limitations section - If the data here were used in conjunction with Brainbow for neuronal labelling, while it is an advantage that existing are line could be used, one would her to make new Brainbow transgenic lines with the selected mutant LoxP sites. This would be a considerable undertaking if several variants were to be tried. Discussion of this point could be added.

With these changes and careful proof reading of the newly added sections for typographical errors, I believe the manuscript is acceptable for publication.

Reviewer #2: (No Response)

Reviewer #3: I appreciate the authors’ great effort to improve this manuscript, especially they tested additional mutant loxP sequences with less than 1 % cleavage rate, thoroughly compared the difference of their research to the previous studies on loxP variances, and at the end, they honestly discussed the limitation of this study. I have no further comment on it, except that the new loxP sequences in Figure 7 are not included in Table 1, please add them.

7. PLOS authors have the option to publish the peer review history of their article (what does this mean?). If published, this will include your full peer review and any attached files.

Reviewer #1: No

Reviewer #2: **Yes: **Erika Knott Reece

Reviewer #3: No

---

## [Author Response · Author response to Decision Letter 1]

8 Oct 2022

Response to Reviewer #1’s comments

Reviewer #1’s comment 1:

The Authors have addressed the substantial points made in my initial review of the manuscript by including a description of mutants with lower recombination efficiency. I think that the mutants identified represent a valuable resource.

Authors’ response 1:

First of all, we appreciate your time and effort in reviewing our revised manuscript and providing your feedback. We are glad that you have appreciated our revised manuscript.

Reviewer #1’s comment 2:

However, I think that that the title still overstates what has been done. Mentioning "a novel sparse labelling strategy" is not appropriate when no labelling has been done. Perhaps "Evaluation of a library of loxP variants with a wide range of recombination efficiencies" or something along those lines is appropriate.

Authors’ response 2:

Thank you for your comment. We agree that “a novel sparse labeling strategy” is overstated, as we did not perform sparse labeling in this paper. We changed the title to “Evaluation of a library of loxP variants with a wide range of recombination efficiencies by Cre.” (line 2 in the revised manuscript)

Reviewer #1’s comment 3:

Also the title of first results section "Strategy for achieving sparseness labeling at the desired rate" should be changed to "Strategy for achieving recombination efficiency at the desired rate" or something like that?

Authors’ response 3:

Thank you for your comment. We changed the title of the first result section to “Strategy for achieving recombination efficiency at the desired rate.” (line 90–91 in the revised manuscript)

Reviewer #1’s comment 4:

Also in the limitations section - If the data here were used in conjunction with Brainbow for neuronal labelling, while it is an advantage that existing are line could be used, one would her to make new Brainbow transgenic lines with the selected mutant LoxP sites. This would be a considerable undertaking if several variants were to be tried. Discussion of this point could be added.

Authors’ response 4:

When combining the mutant loxP sequences with the Brainbow system, we can use existing mouse lines for the Cre expression system. For the Brainbow transgenic lines, we have to establish a new line for each mutant loxP sequence. This experiment requires a lot of labor. There is currently no way to reduce this effort, and it will need to be solved in the future. This discussion is added to the limitation section (lines 343–346 in the revised manuscript).

Reviewer #1’s comment 5:

With these changes and careful proof reading of the newly added sections for typographical errors, I believe the manuscript is acceptable for publication.

Authors’ response 5:

Thank you for your comment. We made corrections to the comments we received and carefully proofread the newly added sections for typographical errors.

 

Response to Reviewer #3’s comments

Reviewer #3’s comment 1:

I appreciate the authors’ great effort to improve this manuscript, especially they tested additional mutant loxP sequences with less than 1 % cleavage rate, thoroughly compared the difference of their research to the previous studies on loxP variances, and at the end, they honestly discussed the limitation of this study. 

Authors’ response 1:

First of all, we appreciate your time and effort in reviewing our revised manuscript and providing your feedback. We are glad that you have appreciated our revised manuscript.

Reviewer #3’s comment 2:

I have no further comment on it, except that the new loxP sequences in Figure 7 are not included in Table 1, please add them.

Authors’ response 2:

Thank you for your comment. Table 1 shows the loxP cleavage rates evaluated by Illumina sequencing ordered from lowest to highest. Cleavage rates of the new mutant loxP sequences in Figure 7 are measured by qPCR. Thus, we listed the cleavage rates of mutant loxP sequences evaluated by qPCR as a new supplement table (S2 Table). With the addition of the new S2 table, the previous S2_table was changed to S3_table. In addition, the corresponding sections of the manuscript were revised respectively (lines 232, 258–259, 351, 415, 596–597 in the revised manuscript).

---

## [Editor Report · Decision Letter 2]

12 Oct 2022

Evaluation of a library of loxP variants with a wide range of recombination efficiencies by Cre

PONE-D-22-18681R2

Dear Dr. Aoki,

We’re pleased to inform you that your manuscript has been judged scientifically suitable for publication and will be formally accepted for publication once it meets all outstanding technical requirements.

Kind regards,

Xiao-Hong Lu, M.D., Ph.D.

Academic Editor

PLOS ONE
---

## [Editor Report · Acceptance letter]

14 Oct 2022

PONE-D-22-18681R2 

Evaluation of a library of *loxP* variants with a wide range of recombination efficiencies by Cre 

Dear Dr. Aoki:

I'm pleased to inform you that your manuscript has been deemed suitable for publication in PLOS ONE. Congratulations! Your manuscript is now with our production department. 

Kind regards, 

on behalf of

Dr. Xiao-Hong Lu 

Academic Editor

PLOS ONE